# Evaluation of Smart Alarm Systems for Industry 4.0 Technologies

**Che-Wei Chang** 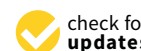

Department of Recreational & Graduate Institute of Recreational Sport Management, National Taiwan University of Sport, Taoyuan City 33301, Taiwan; chewei@gm.ntupes.edu.tw; Tel.: +886-4-22213108

**Abstract:** Traditionally, the footwear industry is labor intensive, and cost control is key to ensuring shoe companies can be competitive. The development of Industry 4.0 concepts, used in high-tech industries and blockchain production information systems, enables the creation of smart factories with online alarm management systems, to improve manufacturing efficiency and reduce human resource requirements. In this paper, the performances of the causal association assessment model and the technique for order preference by similarity to the ideal solution (TOPSIS) model in evaluating large data blockchain technologies and quality online real-time early warning systems for production and raw material supplier management are compared, to increase the intelligence of production and to manage product traceability.

**Keywords:** medium-sized enterprise; collaborative technology; Industry 4.0; intelligent alarm management system (IAMS); cause–effect grey relational analysis (CEGRA); technique for order preference by similarity to the ideal solution (TOPSIS)

## 1. Introduction

With improvements in global living standards, the traditional shoe industry has gradually changed; co-creation design practices have been adopted, allowing the client to dictate the product, while functional sports shoes, casual shoes, and other types of footwear have become essential items with a large consumer base. In this industry, upstream raw materials include textiles, rubber, and plastics, which account for about 60% of the overall cost of the shoe. Different parts of the shoe are produced with different raw materials. For example, the soles of sports shoes are composed of foam rubber, the base consists of an economic value added (EVA) foam model, and the upper portions are composed of leather or mesh. Hence, the shoe production process uses several diverse products, and labor is intensive but low-cost. As a result, it is difficult to employ high-tech manufacturing processes in this industry while reducing raw material costs, improving product quality, and reducing human resource requirements.

In addition, in the footwear industry, a need already exists for more small- and medium-sized factories. However, it is possible that, in the future, only large factories will be able to continue to survive, as based on the appreciation of exchange rates, labor and raw material costs will continue to rise. This makes it difficult for smaller factories to innovate and upgrade, causing them to be uncompetitive. For Industry 4.0 production technologies to be successful in the shoe industry, the human-based corporate culture will also require modification. For instance, if a company does not train talent or allocate resources effectively, the import cost of its product will become too high, complicating the company's transformation and operation. In the case discussed in this study, intelligent online manufacturing and alarm systems from high-tech industry were combined with blockchain production technology, to reduce the demand for human resources and improve the quality of several mass-produced varieties of shoes.

Intelligent alarm management systems (IAMSs) suppress nuisance alarms and provide valuable advisory information to help operators focus quickly on important information and undertake corrective actions [1]. An IAMS provides an expert decision-making system for rapid management of large amounts of data and transformation of data into information, thus reducing human resource requirements and preventing misjudgment and negligence. These systems have been applied to distributed control systems, validated in online tests, and accepted by plant operators. To ensure that a manufacturing process is not interrupted, a real-time IAMS is necessary [2,3]. In addition, since companies require an IAMS that is tailor-made for their operations, they must cooperate with vendors in exchanging information and technology to ensure that the software that is eventually purchased aligns with their business requirements [4,5]. The most effective collaborative technology must select for software contractors [6,7], and the most appropriate and effective approach for evaluating results. Essentially, an IAMS must promote cooperation between enterprises in developing software [8–10]. In this study, the performances of the cause-effect grey relational analysis (CEGRA) [11] and the technique for order preference by similarity to the ideal solution (TOPSIS) [12] models in evaluating the collaborative technology created by IAMS software contractors are compared, to reduce system faults that prevent these products from meeting end users' requirements.

## 2. Background

This section considers the background to the practical work conducted in this study in two parts. In the first, evaluation methodologies for intelligent systems are reviewed, while the second part introduces the CEGRA and TOPSIS algorithms used for decision making in this study.

### 2.1. Review of Evaluation Methodologies for Intelligent Systems

Determining how to increase the product yield and reduce the costs of small- and medium-sized enterprises is a major challenge. In the semiconductor industry, manufacturers have gradually integrated with smaller firms to increase production and reduce costs [13]. Since current manufacturing systems employ 24-hour production cycles, capacities can decrease, producing fewer profits, if the system or a machine malfunctions [14,15]. To help factory operators to quickly undertake measures to prevent accidents, prediction of the risk index is significant. Hence, a number of techniques have been developed for quantifiable risk evaluation and decision making. By combining the concepts of fuzzy set theory, entropy, and ideal and grey relational analysis, Liao et al. developed a fuzzy grey relational method for multiple criteria decision-making problems [16]. Tzeng and Huang [17] proposed a grey relational analysis approach for selecting and reconfiguring aspired-for global manufacturing and logistics systems. In optimization of electrical discharge machining processes, Pradhan [18] used grey relational analysis to estimate the effect of machining parameters on a tool's response, and determined the weights of responses using principal component analysis (PCA), further evaluating the weights of the responses using a response surface methodology. Mondal et al. [19] combined Taguchi and grey relational analysis to evaluate optimal parameters for laser cladding of a steel surface. Sun [20] combined grey relational analysis and information entropy, finding that, in evaluation of the performance of notebook companies, the results obtained with their technique were more objective than those obtained with other decision-making methods. Fayaz et al. [21] proposed a hierarchical fuzzy logic model to assess significant risks of accidents in underground facilities. With their technique, the risk index is predicted using a Kalman filter after a risk is assessed.

Each decision-making method has its advantages and disadvantages, as well as timing and use restrictions. For example, the analytic hierarchy process (AHP) [22] is applied to decision making in uncertain situations where the majority of the evaluation criteria is known. However, the AHP has the following shortcomings: (1) the evaluation scale is a subjective one to nine judgment; (2) the number of elements in the hierarchy should not exceed seven, as it will affect the consistency of the level; and (3) the consistency of the results of analysis is affected by deviation from the selection criteria and evaluation by too large a number of experts. Shih et al. [23] reported that in considering

the benefit or the cost criterion, the TOPSIS method can only reflect the relative proximity of the evaluation criteria within each assessment scheme, and does not reflect the relative proximity to the ideal optimal scheme. Shannon [24] suggested that if data can be obtained directly from the decision-making environment, the entropy algorithm can be used to calculate the objective weight of the criterion. More chaotic measurements increase the weight of this criterion, reflecting larger differences between the importance of different criteria. For such measurements, the entropy method is more effective in managing uncertainty, and the reduction of subjective factors is greater. With CEGRA, both qualitative and quantitative relationships can be identified from complex factors with insufficient information. The main feature of CEGRA is that it can be applied when information is limited, and can support an objective decision based on different data. This method combines the advantages of AHP's expert evaluation, entropy, and TOPSIS, and can be used to effectively manage decision uncertainty, multi-criteria input, and discrete data.

### 2.2. The CEGRA Method

The CEGRA method can use decision-making and information evaluation to incompletely explore the degree of association between two series, quantify the distance between observed and target values using discrete measures, conduct correlation analyses, establish models, and enable decision making between various schemes. CEGRA can effectively manage uncertainty, multivariate input, and discrete data in decision problems [25–27].

In this study, novel algorithms were designed to assess the decision making of an IAMS for the online manufacturing of sports shoes. The implementation of the algorithms used in this study are based on the CEGRA and grey situation decision-making system proposed by Deng [11]. The steps in the algorithms are described as follows:

*Step 1.1: Defining evaluations valuable for assessing online sports shoe online manufacturing IAMSs.*

Let (*EPF*, *X*) be the exploiting resource space when *EPF* is equipped with mapping functions as follows:

$$EPF: \text{Resources} \rightarrow \text{efficacies (v)} \tag{1}$$

In the above, a general consensus among experts has been reached to establish a model, facilitating the ultimate goal of evaluating the performance of collaboration between technology companies according to CEGRA concepts.

*Step 1.2: Categorizing data and evaluating performance.*

Data from the IAMSs can be categorized as data resources or resources possessing information, $\theta(d)$, as shown below:

$$\theta(d) = \{v(\bullet), value; s(\bullet), sign; p(\bullet), polarity; b(\bullet), background; c(\bullet), connotation\}$$

Otherwise, only digits exist, without a resource.

If $dec_o(\bullet)$ is the entirety of complete decision-making information for event *o*, the following data category is obtained:

$$\text{Decision-making data}: \ \theta(d) \in dec_o(\bullet) \tag{2}$$

*Step 1.3: Finding patterns based on grey resource theory.*

In grey resource theory, the situation, $S_{ij} = (a_i, b_j)$, is defined as a pattern, where $a_i$ is a decision maker and $b_j$ is a resource. Hence, the effectiveness of $a_i$ and $b_j$ can expressed using $S_{ij}$.

*GM* is defined as a mapping function, $GM(S_{ij}) = R_{ij}$, where $R_{ij}$ is the value of the mapping function, such that,

$$S_{ij} = (a_i, b_j), S_{ij} = S_{GM} \text{ (grey modeling pattern)} \tag{3}$$

Hence, if the mapping of an event, $E_{ij}$, is defined as $GM(E_{ij}) = R_{ij}$, the synthetic measured effectiveness value for this event is defined as:

$$R_i^{\Sigma} = \frac{1}{n}\sum_{j=1}^{n} R_{ij} \tag{4}$$

*Step 1.4: Contrasting cause–effect incidences and normalizing data.*

In the cause–effect space $(P_{cau}, P_{eff})$, a cause–effect relationship can only be obtained between individuals with cause–effect incidences that are essential contrasts; no contrasts are required for irrelevant individuals. In this study, cause–effect incidence contrasting was defined as follows.

First, let $\alpha$ and $\beta$ be the contrasted individuals, $\alpha, \beta \in P$. $\mu = \Delta = \alpha - \beta$ then represents a comparison gene, and for series comparison, $\Delta = \Delta_{0i}(k) = |\chi_0(k) - \chi_i(k)|$, where $\Delta \to 0$ denotes melt and $\Delta \to \infty$ indicates irrelevance.

The contrasted result, $v$ is given as follows:

$$v = \frac{\varsigma + \pi}{\mu + \pi}, \; \varsigma \to 0 \tag{5}$$

If $(\alpha^{\varsigma}, \beta)$ is considered a cause–effect related contrast, then $\alpha \gg \beta$ or $\beta \ll \alpha \Rightarrow \mu \to \infty$ if $\alpha$ and $\beta$ are irrelevant, such that $v \to 0$. Alternatively, if $\alpha$ and $\beta$ are incident in terms of cause and effect, then $\alpha \to \beta \Rightarrow \mu \to 0 \Rightarrow v \to 1$.

Before calculation of the grey relational coefficients, the data series can be analyzed based on the linearity of normalized data (required to prevent data from being distorted) [28,29], and the following three metrics:

(1)　The upper-bound effectiveness, measured as

$$x_i^*(k) = \frac{x_i(k) - \min\limits_{k} x_i(k)}{\max\limits_{k} x_i(k) - \min\limits_{k} x_i(k)} \tag{6}$$

(2)　The lower-bound effectiveness, measured as

$$x_i^*(k) = \frac{\max\limits_{k} x_i(k) - x_i(k)}{\max\limits_{k} x_i(k) - \min\limits_{k} x_i(k)} \tag{7}$$

(3)　Moderate effectiveness, defined as

If $\min\limits_{k} x_i(k) \le x_{ob}(k) \le \max\limits_{k} x_i(k)$, then:

$$x_i^*(k) = \frac{|x_i(k) - x_{ob}(k)|}{\max\limits_{k} x_i(k) - \min\limits_{k} x_i(k)} \tag{8}$$

If $\max\limits_{k} x_i(k) \le x_{ob}(k)$, then:

$$x_i^*(k) = \frac{x_i(k) - \min\limits_{k} x_i(k)}{x_{ob}(k) - \min\limits_{k} x_i(k)} \tag{9}$$

If $x_{ob}(k) \le \min\limits_{k} x_i(k)$, then:

$$x_i^*(k) = \frac{\max\limits_{k} x_i(k) - x_i(k)}{\max\limits_{k} x_i(k) - x_{ob}(k)} \tag{10}$$

where $x_{ob}(k)$ is the objective value of entity $k$.

*Step 1.5: Examining the architecture of grey relational coefficients.*

In a cause–effect resource incidence space $(P_{cau}, P_{eff})$, the architecture of the grey relational coefficients must enable maximum exploitation of ecological benefits according to cause–effect incidence contrasting and the adjustable coefficient, $\rho$, $\rho \in (0, 1)$. This coefficient must be based on cause and effect criteria with complete incidence. The architecture of grey relational coefficients is expressed as follows:

$$\gamma_{0i}(k) = \gamma(x_0(k), x_i(k)) = \frac{\Delta_{\min} + \rho\Delta_{\max}}{\Delta_{0i}(k) + \rho\Delta_{\max}} \tag{11}$$

The grey relational grade (GRG) for a series $X_i$ is given as follows:

$$\Gamma_{0i} = \sum_{k=1}^{J} w_k\gamma_{0i}(k) \tag{12}$$

where $w_k$ is the weight of the $j$th entity. If no weights need to be applied, then an average is taken, i.e., $\omega_k = \frac{1}{J}$.

*2.3. The TOPSIS Method*

The TOPSIS method for multiple attribute group decision making was developed by Hwang and Yoon [12]. It has subsequently been used to evaluate the purchase of business intelligence systems, product quality improvements, flow control in a manufacturing system, and intelligent home energy management in smart grids [30–33].

In this paper, a combination of the TOPSIS method as introduced by Hwang and Yoon [12], and the implementation described by Dutta et al. [34] is applied for evaluation. The calculation steps are summarized as follows:

*Step 2.1: Establishing an IAMS evaluation decision matrix.*

The IAMS decision matrix, $S$, is defined as below,

$$S = \begin{array}{c} \\ a_1 \\ a_2 \\ \vdots \\ a_i \\ \vdots \\ a_m \end{array} \overset{\begin{pmatrix} b_1 & b_2 & \cdots & b_j & \cdots & b_n \end{pmatrix}}{\begin{bmatrix} x_{11} & x_{12} & \cdots & x_{1j} & \cdots & x_{1n} \\ x_{21} & x_{22} & \cdots & x_{2j} & \cdots & x_{2n} \\ \vdots & \vdots & \cdots & \vdots & \vdots & \vdots \\ x_{i1} & x_{i2} & \vdots & x_{ij} & \vdots & x_{in} \\ \vdots & \vdots & \cdots & \vdots & \vdots & \vdots \\ x_{m1} & x_{m2} & \cdots & x_{mj} & \cdots & x_{mn} \end{bmatrix}} \tag{13}$$

where $a_i$ denotes the evaluated online manufacturing IAMSs, $b_j$ represents the criterion evaluated in *the online manufacturing IAMSs*, $i = 1, 2, ..., m$, $j = 1, 2, ..., n$, and $x_{ij}$ indicates the performance rating of an evaluated online manufacturing IAMS $a_i$ with respect to criterion $b_j$.

*Step 2.2: Data Normalization.*

Data is transformed to a normalized scale as follows:

$$\text{for benefit criteria} \, r_{ij} = \frac{x_{ij}}{\sqrt{\sum\limits_{i=1}^{m} x_{ij}^2}} \tag{14}$$

$$\text{for cost criteria } r_{ij} = \frac{1/x_{ij}}{\sqrt{\sum\limits_{i=1}^{m} 1/x_{ij}^2}} \text{for } i = 1, 2, \ldots, m; j = 1, 2, \ldots, n \tag{15}$$

*Step 2.3: Establishing a weighted normalization matrix*

In the TOPSIS framework, the weighted normalized performance matrix is defined as

$$V = \begin{bmatrix} v_{11} & v_{12} & \cdots & v_{1j} & \cdots & v_{1n} \\ v_{21} & v_{22} & \cdots & v_{2j} & \cdots & v_{2n} \\ \vdots & \vdots & \cdots & \vdots & \vdots & \vdots \\ v_{i1} & v_{i2} & \vdots & v_{ij} & \vdots & v_{in} \\ \vdots & \vdots & \cdots & \vdots & \vdots & \vdots \\ v_{m1} & v_{m2} & \cdots & v_{mj} & \cdots & v_{mn} \end{bmatrix} \tag{16}$$

$$v_{ij} = w_j \times x_{ij}, \text{ for } i = 1, 2, ..., m; \ j = 1, 2, ..., n,$$

where $w_j$ denotes the weight of criterion $j$.

*Step 2.4: Calculating the separation measures*

To calculate the separation measures, which, in this study, characterize an IAMS's proximity to optimal performance, the ideal solution is first calculated as follows

$$d^+ = (v_1^+, v_2^+, \cdots, v_n^+), \text{ where } v_j^+ = \max_i v_{ij} \tag{17}$$

$$d^- = (v_1^-, v_2^-, \cdots, v_n^-), \text{ where } v_j^- = \min_i v_{ij} \tag{18}$$

The distance between the ideal solution and the negative ideal solution for each alternative is then calculated as

$$d_i^+ = \sqrt{\sum_{j=1}^{n} \left( v_{ij} - v_j^+ \right)^2} \quad i = 1, 2, \ldots, m, \tag{19}$$

$$d_i^- = \sqrt{\sum_{j=1}^{n} \left( v_{ij} - v_j^- \right)^2} \quad i = 1, 2, \ldots, m, \tag{20}$$

*Step 2.5: Calculating the relative closeness coefficient and rank the center preference order*

The relative closeness to the ideal solution of each online manufacturing IAMSs can be calculated as:

$$a_i^* = \frac{d_i^-}{d_i^+ + d_i^-}, \text{ for } i = 1, 2, \ldots, m \tag{21}$$

## 3. Case Study

The Datong Shoes and Materials Company was founded in 1992 with USD $1,000,000 of capital. After the 2008 global financial crisis, their revenue declined annually and their gross profit margin turned negative; the more orders received, the larger the losses they made. Thus, the company decided to reduce its operating costs, investing in new technologies. In 2013, after introducing IAMS and blockchain production information systems, the business began to grow steadily, with turnover averaging around USD $3,500,000. With the introduction of automation, manpower was no longer the most urgent requirement. Hence, the number of factory production line workers decreased from about

1200 in 2013, to 280 by the end of 2018, which is an annual reduction of 25% in human resources. The gross profit margin grew rapidly from 8.61% to 36.28% in 2018 as shown in Table 1.

**Table 1.** Turnover and gross profit margin of Datong Shoe and Materials Company, 2013–2018. (USD).

|  | 2013 | 2014 | 2015 | 2016 | 2017 | 2018 |
|---|---|---|---|---|---|---|
| Turnover | $3,483,534 | $3,968,090 | $3,645,004 | $3,624,018 | $3,572,838 | $3,763,531 |
| Gross Margin | 8.61% | 16.89% | 35.75% | 33.68% | 30.45% | 36.28% |

The company has a corporate culture that supports innovation, believing that exclusive companies import automated intelligent factory technologies, and that a wealth of manufacturing experience is an important asset for the sustainable development of a firm. In terms of demographics, 70% of the engineers at the Datong Shoe Material Company are over 50 years old. Therefore, in the company's position, developing an artificial intelligence system as soon as possible is essential for transferring inherited knowledge from generation to generation, to preserve the knowledge of the older employees. At the same time, future generations should inherit the company's goals of innovative development and sustainable management. Investment in new technologies should include initial input machine data exchanges, the online cloud, automated operations, big data analysis, and other related technologies [35]. This requires the investment of a large amount of capital and talent training and transformation to integrate the existing manufacturing technology, quality, and sales and product experience to establish an adaptable and resource-efficient intelligent plant. In terms of business and value processes, customer and business partner services must be integrated to provide excellent after-sales service. The company plans to have 100 staff and 100 employees for operation in 2019, to minimize its human resource requirements. From the viewpoint of turnover, as shown in Table 1, the goal is to maintain the same level of substantial growth of about 30%, which before was included as part of the gross margin, not the revenue.

Four collaborative technology software companies in Taiwan that specialize in code development were assessed in this study. Company A is Data Systems Consulting Co., Ltd. (Taichung, Taiwan), which specializes in ERP and customized software. Company B is IBM Co., Ltd., which specializes in information management systems and customized software. Company C is Elements Innovation Co., Ltd., which specializes in ISO databases, CRM, and customized software. Company D is Tien Kang Co., Ltd., which specializes in CPA and BI decision support systems. The CEGRA and TOPSIS algorithms for assessing online sports shoe manufacturing IAMSs are summarized in Figure 1.

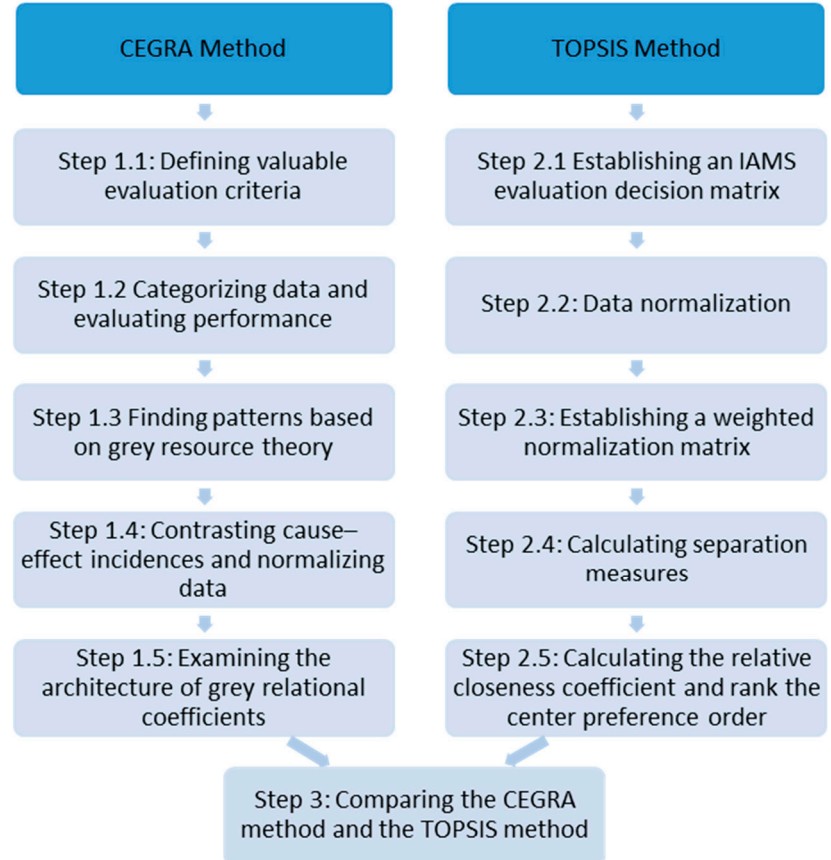

**Figure 1.** Comparing the CEGRA method and the TOPSIS method.

### 3.1. Decision Making Using the CEGRA Method

#### Step 1.1: Defining Valuable Evaluation Criteria

Since a general consensus was reached among experts about the method followed to establish a model, the ultimate goal of evaluating the performance of collaborative technology companies according to CEGRA concepts can be achieved. In this study, 17 evaluation criteria were determined, as summarized in Table 2 [36–38]. Here, larger coefficients for the market share (MS), improving performance and enhancing morale (IPE), performance reward system (PRS), technology and engineering (TE), economies of scope (ES), surge capacity (SC), information sharing (IS), IT capability (ITC), and experience in similar products (ESP) criteria, represent better performance. For all other criteria, smaller coefficients are better.

**Table 2.** Evaluation criteria and system definition or requirement.

| Evaluation Criteria | System Definition or Requirement |
| --- | --- |
| Market share (MS) [39] | Data related to consumer interaction that is added through the Internet of Things, to analyze the market share for sports shoes. |
| Improving performance and enhancing morale (IPE) [40] | Provision of tools, such as information systems, that assist company executives in the management of cross-team projects, reduction of errors, and improvement of job performance and morale. |
| Performance reward system (PRS) [41] | Rewards should be based on accomplishments and performance, and applied consistently across the company. In small business, performance is measured as a combination of attainment in productivity, efficiency, and training. The main goal of rewards is to motivate employees with tangible reasons to continue to improve performance and help the company grow. |

**Table 2.** *Cont.*

| Evaluation Criteria | System Definition or Requirement |
|---|---|
| Technology and engineering (TE) | Demand for new sports shoes may change during or after they are in development. The system should allow for instant customer and engineering changes and outsourced technology. This metric assesses the provisions made for such a response. |
| Economies of scope (ES) [42] | The ability to exploit commonalities in processing for cost-effective operation. For instance, by sharing the same raw materials in the production process, a company can produce two different kinds of sports shoes simultaneously. The approximate cost of each kind of shoe is consequently lower than the average cost required to produce the two types of shoes independently. |
| Surge capacity (SC) [43] | The ability to meet sudden, unexpected increases in demand by expanding production with existing personnel and equipment. |
| Information sharing (IS) [44] | Information sharing and quality are not impacted by customer or technological uncertainty, commitment of supply chain partners, or information technology (IT) enablers. |
| IT capability (ITC) [45] | The organization's ability to identify how IT can meet business needs, use IT to cost-effectively improve business processes, and provide long-term maintenance and support for IT-based systems. |
| Delivery performance (DP) [46] | The customer expectation level met by products and services supplied by an organization. It indicates the potential for the supply chain to provide products and services to customers. |
| Flexibility in billing and payment (FBP) [38] | Conditions that increase goodwill between the partner and the raw material supplier. |
| Quality management (QM) [47] | The relationships between the quality management practices and various levels of organizational performance. |
| Reducing the development and maintenance costs of information systems (RD) [48] | This refers to all recurring sustenance, operations, and maintenance costs accumulated in the planned product life cycle, (in the shoe industry, this encompasses a 20-year span). The system includes documentation, data, hardware, software applications, replacement training, and systems and project management. |
| Human Information Processing (HIP) [49] | Activation of a learned sequence in the long-term memory of elements needed to manufacture sneakers. This is initiated through appropriate inputs and should proceed automatically, without stressing the capacity limitations of the system and without demanding attention or management. |
| Product management (PM) [50] | This metric assesses the institutionalization of methods in the sneaker manufacturing process, to improve the success rate of projects in terms of schedule predictability, quality, and project duration. |
| Experience in similar products (ESP) [51] | Application of previous interactions with customers, detailed examination of records of athletes or consumers in different sports, and of different ages and abilities, or knowledge of previous use and designs, to help clarify consumers' requirements for better functional shoes. |
| More reliable information system procurement and performance (MR) [52] | The footwear industry requires the procurement of several diverse raw materials, which is affected by fluctuations in international futures. This metric assesses the ability to plan for such fluctuations so that purchase prices will be less than half the sale price. |
| Learning new software management technology (LN) [53] | Company employees use of new production technology software to learn new abilities and adapt to different tasks, and scholars are regarded as affecting the company's competitiveness. |

*Step 1.2: Categorizing Data and Evaluating Performance*

Ten experienced members of research and development teams in the sports shoe manufacturing industry were selected as experts to evaluate the modules used in this study with respect to the criteria identified above. In this evaluation, the grey modeling pattern was used as the decision maker, on the

basis of Equations (1) and (2). Table 3 lists the average results of evaluation of each different system in relation to each different criterion.

**Table 3.** The decision-making matrix.

| Company | MS | IPE | PRS | TE | ES | SC | IS | ITC | DP | FBP | QM | RD | HIP | PM | ESP | MR | LN |
|---|---|---|---|---|---|---|---|---|---|---|---|---|---|---|---|---|---|
| Company A | 2.98 | 2.71 | 3.35 | 2.30 | 2.33 | 2.21 | 1.72 | 2.09 | 2.04 | 2.25 | 1.91 | 2.17 | 2.16 | 2.64 | 2.08 | 2.04 | 1.96 |
| Company B | 2.19 | 2.32 | 2.08 | 2.62 | 2.62 | 2.64 | 3.39 | 2.36 | 2.51 | 2.43 | 2.65 | 2.65 | 2.54 | 2.31 | 2.39 | 2.72 | 2.34 |
| Company C | 2.35 | 2.29 | 2.04 | 2.65 | 2.69 | 2.59 | 2.61 | 3.06 | 3.04 | 2.59 | 3.05 | 2.62 | 2.62 | 2.60 | 2.61 | 2.71 | 2.62 |
| Company D | 2.49 | 2.68 | 2.53 | 2.43 | 2.37 | 2.56 | 2.28 | 2.49 | 2.41 | 2.74 | 2.39 | 2.56 | 2.68 | 2.44 | 2.91 | 2.53 | 3.09 |

*Step 1.3: Finding Patterns Based on Grey Resource Theory* The criteria for contrasting individuals

representing a comparison gene were set as defined by Equations (3) and (4). In the resource cause–effect incidence space, the architecture of the grey relational coefficients was consistent with contrasting cause–effect incidences and the symbols for "top scale," "button scale," and "adjustable coefficient," and was based on the complete incidence of cause–effect criteria.

*Step 1.4: Contrasting Cause–Effect Incidences and Normalizing Data*

Based on previous definitions, the reference series in this study is $X_0$ = (2.98, 2.71, 3.35, 2.65, 2.69, 2.64, 3.39, 3.06, 2.04, 2.25, 1.91, 2.17, 2.16, 2.31, 2.91, 2.04, and 1.96). Data were normalized to this series using Equations (6) and (7), and the relational coefficients were then computed, as summarized in Table 4.

**Table 4.** Summaries of the resultant relational coefficients.

| | MS | IPE | PR | TE | ES | SC | IS | ITC | DP | FBP | QM | RD | HIP | PM | ESP | MR | LN | $\Gamma_{0i}$ | Rank |
|---|---|---|---|---|---|---|---|---|---|---|---|---|---|---|---|---|---|---|---|
| Company A | 1.00 | 1.00 | 1.00 | 0.50 | 0.50 | 0.50 | 0.50 | 0.50 | 1.00 | 1.00 | 1.00 | 1.00 | 1.00 | 0.50 | 0.50 | 1.00 | 1.00 | 0.7941 | 1 |
| Company B | 0.50 | 0.52 | 0.51 | 0.92 | 0.84 | 1.00 | 1.00 | 0.58 | 0.68 | 0.73 | 0.61 | 0.50 | 0.58 | 1.00 | 0.61 | 0.50 | 0.75 | 0.6959 | 2 |
| Company C | 0.56 | 0.50 | 0.50 | 1.00 | 1.00 | 0.90 | 0.68 | 1.00 | 0.50 | 0.59 | 0.50 | 0.52 | 0.53 | 0.53 | 0.73 | 0.50 | 0.63 | 0.6571 | 3 |
| Company D | 0.62 | 0.93 | 0.62 | 0.61 | 0.53 | 0.84 | 0.60 | 0.63 | 0.73 | 0.50 | 0.70 | 0.55 | 0.50 | 0.72 | 1.00 | 0.58 | 0.50 | 0.6565 | 4 |

*Step 1.5: Examining the Architecture of Grey Relational Coefficients*

The $\gamma_{0i}(k)$ series was computed using Equation (11), following a comparison between $X_0$ and the relational coefficients. GRG weights were subsequently calculated using Equation (12), with the results summarized in the right-hand column of Table 4. The performance values of companies A, B, C, and D were 0.79, 0.70, 0.66, and 0.66, respectively. Hence, Company A has the best performance, based on CEGRA evaluation. As it sells products at the most reasonable price and is a reputable software vendor, it can help operators in the sports shoe industry understand warning messages more clearly.

*3.2. Decision Making Using the TOPSIS Method*

*Step 2.1: Establishing an IAMS evaluation decision matrix*

The decision matrix for evaluating the IAMSs was established as described by Equation (13), using the data in Table 3.

*Step 2.2: Data Normalization*

As with the CEGRA, large coefficients for MS, IPE, PR, TE, ES, SC, IS, ITC, QM, and ESP, represent good performance, while smaller coefficients represent good performance for DP, FBP, RD, HIP, PM, MR and LN. The former 10 metrics were categorized as benefit criteria in the TOPSIS framework, while the latter 7 were categorized as cost criteria. Hence, data were normalized using Equation (14) or (15), as appropriate, with results as shown in Table 5.

**Table 5.** Normalization of evaluation IAMSs decision matrix.

| Company | MS | IPE | PRS | TE | ES | SC | IS | ITC | DP | FBP | QM | RD | HIP | PM | ESP | MR | LN |
|---|---|---|---|---|---|---|---|---|---|---|---|---|---|---|---|---|---|
| Company A | 0.5913 | 0.5403 | 0.6556 | 0.4592 | 0.4646 | 0.4410 | 0.3344 | 0.4139 | 0.7720 | 0.8181 | 0.3769 | 0.9810 | 0.9597 | 0.0000 | 0.4134 | 0.9630 | 0.7873 |
| Company B | 0.4345 | 0.4626 | 0.4071 | 0.5231 | 0.5225 | 0.5268 | 0.6590 | 0.4673 | 0.4092 | 0.5176 | 0.5229 | 0.0000 | 0.2584 | 0.8506 | 0.4750 | 0.0000 | 0.5225 |
| Company C | 0.4663 | 0.4566 | 0.3992 | 0.5291 | 0.5364 | 0.5168 | 0.5074 | 0.6060 | 0.0000 | 0.2505 | 0.6018 | 0.0613 | 0.1107 | 0.1031 | 0.5187 | 0.0142 | 0.3274 |
| Company D | 0.4941 | 0.5344 | 0.4951 | 0.4852 | 0.4726 | 0.5108 | 0.4432 | 0.4931 | 0.4864 | 0.0000 | 0.4716 | 0.1839 | 0.0000 | 0.5155 | 0.5783 | 0.2691 | 0.0000 |

### Step 2.3: Establishing a weighted normalization matrix

The weighted normalized decision matrix depicted in Table 6 was calculated as described by Equation (16). In this research, it was assumed that all evaluation criteria had the same weight.

**Table 6.** Weighted normalization IAMS matrix.

| Company | MS | IPE | PRS | TE | ES | SC | IS | ITC | DP | FBP | QM | RD | HIP | PM | ESP | MR | LN |
|---|---|---|---|---|---|---|---|---|---|---|---|---|---|---|---|---|---|
| Company A | 0.0348 | 0.0318 | 0.0386 | 0.0270 | 0.0273 | 0.0259 | 0.0197 | 0.0243 | 0.0454 | 0.0481 | 0.0222 | 0.0577 | 0.0565 | 0.0000 | 0.0243 | 0.0566 | 0.0463 |
| Company B | 0.0256 | 0.0272 | 0.0239 | 0.0308 | 0.0307 | 0.0310 | 0.0388 | 0.0275 | 0.0241 | 0.0304 | 0.0308 | 0.0000 | 0.0152 | 0.0500 | 0.0279 | 0.0000 | 0.0307 |
| Company C | 0.0274 | 0.0269 | 0.0235 | 0.0311 | 0.0316 | 0.0304 | 0.0298 | 0.0356 | 0.0000 | 0.0147 | 0.0354 | 0.0036 | 0.0065 | 0.0061 | 0.0305 | 0.0008 | 0.0193 |
| Company D | 0.0291 | 0.0314 | 0.0291 | 0.0285 | 0.0278 | 0.0300 | 0.0261 | 0.0290 | 0.0286 | 0.0000 | 0.0277 | 0.0108 | 0.0000 | 0.0303 | 0.0340 | 0.0158 | 0.0000 |

### Step 2.4: Calculating the separation measures

The separation of the benefit criteria and cost criteria of the four companies from the ideal solution were calculated using Equations (17) and (18). Results are as shown in Tables 7 and 8.

**Table 7.** The distance between the ideal solution.

| Company | MS | IPE | PRS | TE | ES | SC | IS | ITC | DP | FBP | QM | RD | HIP | PM | ESP | MR | LN |
|---|---|---|---|---|---|---|---|---|---|---|---|---|---|---|---|---|---|
| Company A | 0.0000 | 0.0000 | 0.0000 | 0.0041 | 0.0042 | 0.0050 | 0.0191 | 0.0113 | 0.0000 | 0.0000 | 0.0132 | 0.0000 | 0.0000 | 0.0500 | 0.0097 | 0.0000 | 0.0000 |
| Company B | 0.0092 | 0.0046 | 0.0146 | 0.0004 | 0.0008 | 0.0000 | 0.0000 | 0.0082 | 0.0213 | 0.0177 | 0.0046 | 0.0577 | 0.0413 | 0.0000 | 0.0061 | 0.0566 | 0.0156 |
| Company C | 0.0074 | 0.0049 | 0.0151 | 0.0000 | 0.0000 | 0.0006 | 0.0089 | 0.0000 | 0.0454 | 0.0334 | 0.0000 | 0.0541 | 0.0499 | 0.0440 | 0.0035 | 0.0558 | 0.0270 |
| Company D | 0.0057 | 0.0004 | 0.0094 | 0.0026 | 0.0038 | 0.0009 | 0.0127 | 0.0066 | 0.0168 | 0.0481 | 0.0077 | 0.0469 | 0.0565 | 0.0197 | 0.0000 | 0.0408 | 0.0463 |

**Table 8.** The distance between the negative solution.

| Company | MS | IPE | PRS | TE | ES | SC | IS | ITC | DP | FBP | QM | RD | HIP | PM | ESP | MR | LN |
|---|---|---|---|---|---|---|---|---|---|---|---|---|---|---|---|---|---|
| Company A | 0.0092 | 0.0049 | 0.0151 | 0.0000 | 0.0000 | 0.0000 | 0.0000 | 0.0000 | 0.0454 | 0.0481 | 0.0000 | 0.0577 | 0.0565 | 0.0000 | 0.0000 | 0.0566 | 0.0463 |
| Company B | 0.0000 | 0.0004 | 0.0005 | 0.0038 | 0.0034 | 0.0050 | 0.0191 | 0.0031 | 0.0241 | 0.0304 | 0.0086 | 0.0000 | 0.0152 | 0.0500 | 0.0036 | 0.0000 | 0.0307 |
| Company C | 0.0019 | 0.0000 | 0.0000 | 0.0041 | 0.0042 | 0.0045 | 0.0102 | 0.0113 | 0.0000 | 0.0147 | 0.0132 | 0.0036 | 0.0065 | 0.0061 | 0.0062 | 0.0008 | 0.0193 |
| Company D | 0.0035 | 0.0046 | 0.0056 | 0.0015 | 0.0005 | 0.0041 | 0.0064 | 0.0047 | 0.0286 | 0.0000 | 0.0056 | 0.0108 | 0.0000 | 0.0303 | 0.0097 | 0.0158 | 0.0000 |

### Step 2.5: Calculating the relative closeness coefficient and rank the center preference order

Using the separation measures summarized above, the relative closeness coefficient, $a_i^*$, for the four companies, was calculated based on Equation (21), as shown in the TOPSIS performance values column of Table 9. Each company was subsequently ranked in order of preference based on this metric.

**Table 9.** Outcome of the TOPSIS.

| | TOPSIS Ideal Solution $d^+$ | TOPSIS Negative Ideal Solution $d^-$ | TOPSIS Performance Values | Rank |
|---|---|---|---|---|
| Company A | 0.0577 | 0.1060 | 0.6476 | 1 |
| Company B | 0.0790 | 0.0690 | 0.4659 | 2 |
| Company C | 0.1045 | 0.0285 | 0.2142 | 4 |
| Company D | 0.0937 | 0.0461 | 0.3299 | 3 |

### Step 3: Comparing the CEGRA Method and the TOPSIS Method

The results of evaluation using the CEGRA and TOPSIS algorithms are given in the right-most column of Tables 4 and 9, respectively. A comparison of these columns shows that the first- and second-ranked results were the same for both methods. However, the CEGRA method is simpler to implement than TOPSIS, highlighting its potential for use by decision makers with limited technical ability.

## 4. Conclusions

The traditional footwear industry is labor-intensive, and companies' profits are constantly compressed as a result of rising wages. Hence, for sustainable operation, the Datong Shoes and Materials Company introduced IAMS and big data analysis systems to become an intelligent factory, greatly reducing their demand for manpower. In this study, we developed a CEGRA model for evaluation of intelligent system suppliers being considered for future collaboration, for further improvement of the company's operational efficiency. In constructing this model, criteria for evaluation of these systems were identified using the focus group discussion method. The CEGRA model for evaluating collaborative technology can be divided into five steps, with different aspects of the IAMS evaluated for quality control. Thus, the model can be used to identify the criteria that most affects the quality of an IAMS, thereby reducing system faults that prevent collaborative technology software products from meeting end users' requirements. As a result, the model enables traditional companies to evaluate the solutions provided by intelligent system suppliers when they may otherwise lack the expertise to do so. Using this model, the most suitable alternative software solution was identified from a list of four candidates.

In addition, a comparison of the findings obtained using the CEGRA model with those obtained from the traditional TOPSIS method, showed that both techniques identified the same options as the top two ranking systems. However, the CEGRA algorithm proposed can manage the uncertainty of decision problems, multiple decision evaluation attributes, and discrete data effectively, providing a simple and convenient method for decision making. Although the model introduced in this study was limited to the introduction of IAMS to upgrade a traditional shoe company to a technology shoe company, the results demonstrate that in future research, CEGRA evaluation models can help decision makers in different traditional industries find suitable solutions for upgrading to technology industries. The evaluation criteria and decision-making processes can provide system integrators with the ability to meet the needs of upgrade vendors, providing better system integration services when developing smart factories.

**Author Contributions:** C.-W.C. conceived designed, analyzed the data, and wrote the paper. All authors have read and agreed to the published version of the manuscript.

**Funding:** This research received no external funding.

**Acknowledgments:** I thank the Datong Shoes and Materials Company for providing the case study information.

**Conflicts of Interest:** The authors declare no conflict of interest.

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
