# Peer review of "Evaluation of Smart Alarm Systems for Industry 4.0 Technologies"

_applsci, doi:10.3390/app10062022_

Round 1
Reviewer 1 Report
Overall a well-organized and relevant study for implementing of smart systems in manufacturing industries!
The presented study improved a lot compared to the previous version! Some modifications are still needed before publishing this work. See below some suggestions:
- As there is a strong focus on "alarm systems": please use this aspect in the title as it makes the study very specific, so the reader should be aware of what is discussed.
- Line 10-13: modify sentence. "...is required..." --- to -- "...aims to improve manufacturing efficiency and can reduce....."
- Line 24: is tis true?? For example, Nike is offering co-creation for some time already, changing the way of offering shoes (client designs own shoes)... Please revise
- Line 41: how does relate Blockchain to this argument?? Please explain briefly or omit.
- Line 254: clear figure, which highlights well the analysis of the paper! Nice work.
- Line 276: Revise grammar: ...In future research...
- Section 4: Please add some lines as recommendation. E.g. how cold this study help other business cases and which are issues??
Author Response
Review 1
- As there is a strong focus on "alarm systems": please use this aspect in the title as it makes the study very specific, so the reader should be aware of what is discussed.
Reply:
Thank you for this suggestion. I have modified the title to “Evaluation of Smart Alarm Systems for Industry 4.0 Technologies ”.
- Line 10-13: modify sentence. "...is required..." --- to -- "...aims to improve manufacturing efficiency and can reduce....."
Reply:
Thank you for your valuable comment. I have rephrased the sentence beginning in Line 10 as follows: “The development of Industry 4.0 concepts, used in high-tech industries and blockchain production information systems, enables the creation of smart factories with online alarm management systems, to improve manufacturing efficiency and reduce human resource requirements.”
- Line 24: is tis true?? For example, Nike is offering co-creation for some time already, changing the way of offering shoes (client designs own shoes)... Please revise
Reply:
Thank you for your valuable comment. I have included a consideration of co-creation practices in the revised version of the sentence beginning on line 25: “With improvements in global living standards, the traditional shoe industry has gradually changed; co-creation design practices have been adopted allowing the client to dictate the product, while functional sports shoes, casual shoes, and other types of footwear have become essential items with a large consumer base.”
- Line 41: how does relate Blockchain to this argument?? Please explain briefly or omit.
Reply:
Thank you for your valuable comment. I have omitted Blockchain from this argument.
- Line 254: clear figure, which highlights well the analysis of the paper! Nice work.
Reply:
Thank you for your this comment.
- Line 276: Revise grammar: ...In future research...
Reply:
Thank you for this correction. I have included a comma in line 391, such that the affected sentence appears as follows: “…in future research, …”.
- Section 4: Please add some lines as recommendation. E.g. how could this study help other business cases and which are issues??
Reply:
Thank you for your valuable comment. I have added the following in Section 4: “The evaluation criteria and decision-making processes can provide system integrators with the ability to meet the needs of upgrade vendors, providing better system integration services when developing smart factories.”

Reviewer 2 Report
Major comments:
- the title of the paper is different from the content as a reader expects an evaluation of all the usually known I4.0 technologies (usually the nine key technologies: Autonomous robots, Simulation, AR, Data Integration, Cloud, Big Data Analytics, Cyber Security, Industrial IoT)
- title should be something like "Evaluation of Intelligent Alarm Management Systems" or talking about the comparison of CEGRA and TOPSIS
Minor comments:
- keywords: "Intelligent" is missing for IAMS
Author Response
Review 2
- the title of the paper is different from the content as a reader expects an evaluation of all the usually known I4.0 technologies (usually the nine key technologies: Autonomous robots, Simulation, AR, Data Integration, Cloud, Big Data Analytics, Cyber Security, Industrial IoT)
title should be something like "Evaluation of Intelligent Alarm Management Systems" or talking about the comparison of CEGRA and TOPSIS
Reply:
Thank you for this suggestion. I have revised the title to “Evaluation of Smart Alarm Systems for Industry 4.0 Technologies”
- - keywords: "Intelligent" is missing for IAMS
Reply:
Thank you for this correction. I have included this revision in the keywords section: “intelligent alarm management system (IAMS)”

Reviewer 3 Report
- The aim is clear is study is not clear. Moreover, it is not clear what the study found or how they found it: this information is not easily identifiable from abstract, title and reference sections
Author Response
Review 3
- The aim is clear is study is not clear. Moreover, it is not clear what the study found or how they found it: this information is not easily identifiable from abstract, title and reference sections
Reply:
Thank you for your valuable comment. I have revised the abstract and title, as below, to address this criticism. In addition, I have created a new “background” section including a review of evaluation methodologies for intelligent systems, to help clarify the relevance of the references to the main aim of the paper.
Abstract:
Traditionally, the footwear industry is labor intensive, and cost control is key to ensuring shoe companies can be competitive. The development of Industry 4.0 concepts, used in high-tech industries and blockchain production information systems, enables the creation of smart factories with online alarm management systems, to improve manufacturing efficiency and reduce human resource requirements. In this paper, the performances of the causal association assessment model and the technique for order preference by similarity to the ideal solution (TOPSIS) model in evaluating large data blockchain technologies and quality online real-time early warning systems for production and raw material supplier management, are compared, to increase the intelligence of production, and to manage product traceability.
Title: Evaluation of Smart Alarm Systems for Industry 4.0 Technologies

This manuscript is a resubmission of an earlier submission. The following is a list of the peer review reports and author responses from that submission.
Round 1
Reviewer 1 Report
Connecting Engineering and Entrepreneurship is very actual, your paper represents this interest.
Paper is very good organised, starting from Abstract, Introductions, Methods and results including Conclusions based on represented results.
Author Response
Point 1: I don't feel qualified to judge about the English language and style.
Response 1: This problem has been modified. This paper has been shortened and modified. Our colleague, who is a native English speaker, has revised the manuscript for syntax and writing style errors.
Point 2: Connecting Engineering and Entrepreneurship is very actual, your paper represents this interest. Paper is very good organised, starting from Abstract, Introductions, Methods and results including Conclusions based on represented results.
Response 2: Thank you for your valuable comments.

Reviewer 2 Report
The paper applied the cause-effect grey relational analysis (CERGA) model for the evaluation of companies providing software solutions to the sport shoe industry.
Firstly, the methodology (Section 2) is not well-written, i.e. no explanation of variables or the meaning of formulation. It is difficult for general readers (with basic background in mathematics) to understand the paper as not everybody would know the basics of grey theory.
Secondly, the title might cause confusing. I think 'sustainability' is the attractive keyword, however, it is hardly for me to link any outcomes of this paper to it. Similarly, 'Industry 4.0' is quite fuzzy. I guess it is related to smart technologies. Therefore, the title is too broad which the content of paper cannot cover all the aspects.
Finally, about the contribution of this paper, I think it is thin, i.e., the propose of 17 evaluation criteria. The main method is CERGA which is introduced in the literature.
Author Response
Point 1: English language and style are fine/minor spell check required
Response 1: This problem has been modified. This paper has been shortened and modified. Our colleague, who is a native English speaker, has revised the manuscript for syntax and writing style errors.
Point 2: The methodology (Section 2) is not well-written, i.e. no explanation of variables or the meaning of formulation. It is difficult for general readers (with basic background in mathematics) to understand the paper as not everybody would know the basics of grey theory.
Response 2: The CEGRA method can use the information of decision-making and evaluation to incompletely explore the degree of association between two series, use discrete measures to measure its distance, and make correlation analysis, model establishment, and decision-making between decision-making schemes. CEGRA can effectively deal with uncertainty, multivariate input, and discrete data of decision problems [42-44].
Point 3: The title might cause confusing. I think 'sustainability' is the attractive keyword, however, it is hardly for me to link any outcomes of this paper to it. Similarly, 'Industry 4.0' is quite fuzzy. I guess it is related to smart technologies. Therefore, the title is too broad which the content of paper cannot cover all the aspects.
Response 3: I revised the title to “Evaluate Industry 4.0 Technologies for Smart Intelligent Systems”.
Point 4: Finally, about the contribution of this paper, I think it is thin, i.e., the propose of 17 evaluation criteria. The main method is CERGA which is introduced in the literature.
Response 4: Thank you for your valuable comments. Table 2 shows the collation of 17 evaluation criteria.

Reviewer 3 Report
An interesting study on proposing intelligent technologies/systems in traditional manufacturing industries, which is (in the light of Industry 4.0) a highly relevant topic!
Before publication I recommend strongly a revision of the manuscript:
revise English grammar and sentences (revise title!) improve justification/argumentation for statements (such as in line 39-42 and 42-45). Why are import costs and operating become more difficult when introducing Industry 4.0 technologies in the footwear industry??? In the introduction section: how are the solar industry and footwear industry related? Not clear!! In section 3 : Justify better with arguments the statement made in line 167-170 Improve the conclusion section (link it better to your case study)Author Response
Point 1: revise English grammar and sentences (revise title!)
Response 1: This problem has been modified. This paper has been shortened and modified. Our colleague, who is a native English speaker, has revised the manuscript for syntax and writing style errors. I revise the title to “Evaluate Industry 4.0 Technologies for Smart Intelligent Systems”.
Point 2: Improve justification/argumentation for statements (such as in line 39-42 and 42-45). Why are import costs and operating become more difficult when introducing Industry 4.0 technologies in the footwear industry???.
Response 2: I rewrote lines 39–42 and 42–45: “In the shoe industry, if the production technology of Industry 4.0 and blockchain is introduced, it will be a human-based corporate culture that impacts the traditional industry. If the company does not do a good job in talent training and company resource allocation, the import cost will be too high, which will make the company difficult to transform and increase the difficulty of the company's operation. Therefore, the case discussed in this study draws on the intelligent online manufacturing and alarm system of high-tech industry, combined with blockchain production technology, to reduce the demand for manpower and improve the quality of several mass-produced varieties of sneakers.”
Point 3: The title might cause confusing. I think 'sustainability' is the attractive keyword, however, it is hardly for me to link any outcomes of this paper to it. Similarly, 'Industry 4.0' is quite fuzzy. I guess it is related to smart technologies. Therefore, the title is too broad which the content of paper cannot cover all the aspects.
Response 3: I added “In the solar and semiconductor industry, manufacturers have gradually been integrated with smaller firms to increase production and reduce costs.” Manufacturers of the footwear industry are also gradually merging small manufacturers. The solar and semiconductor industries are produced 24 hours a day. The fast and smart factory of the shoe industry to be built is also produced 24 hours a day. Industry problems are consistent.
Point 4: In section 3: Justify better with arguments the statement made in line 167-170 Improve the conclusion section
Response 4: I rewrote lines 167–170: “Therefore, in the company's position, it is necessary to develop an artificial intelligence system as soon as possible to preserve the knowledge of the old employees, and to carry on the inheritance from generation to generation, which is the direction of the company's innovation and sustainable development., and at the same time future generations should inherit the company’s moves towards innovative development and sustainable management. Investment would be in new technologies, including initial input machine data exchanges, the online cloud, automated operations, big data analysis, and other related technologies [23].”

Round 2
Reviewer 2 Report
Thank you for the revised work. From my point of view, the methodology is still not well-written. For example, why the concept of 'unweighted justice pattern' is related to the problem. What is the meaning of S_de, S_rat, S_aut, etc? In the Eq (5), what is pi? etc.
Author mentioned about the novel algorithm for assessing the "IAMS decision-making of the new CEGRA in the online manufacture of sports shoes". Please give more details about what is novel compared to which method? The novel method is better than the conventional method in which aspect? What is the assumption that makes the benefit or advantage of the proposed method (or why we should go with CEGRA but not others)?
Author Response
Response to Reviewer 2 Comments
Point 1:
Thank you for the revised work. From my point of view, the methodology is still not well-written. For example, why the concept of 'unweighted justice pattern' is related to the problem. What is the meaning of S_de, S_rat, S_aut, etc? In the Eq (5), what is pi? etc.
Response 1:
I revised Lines 117 to 155 to illustrate CEGRA steps.
Point 2:
Author mentioned about the novel algorithm for assessing the "IAMS decision-making of the new CEGRA in the online manufacture of sports shoes". Please give more details about what is novel compared to which method? The novel method is better than the conventional method in which aspect? What is the assumption that makes the benefit or advantage of the proposed method (or why we should go with CEGRA but not others)?
Response 2:
From lines 79 to 99, I added references [45-47] to illustrate why I use CEGRA method. “Each decision-making method has its own advantages and disadvantages, and it also has the timing and use restrictions. There is no good or bad, and the timing is appropriate. For example: The Analytic Hierarchy Process [45], the main timing is in uncertain situations and decision-making issues with a majority of evaluation criteria. However, there are also shortcomings. (1) The evaluation scale is a sensation of 1~9 judgment. (2) It is best not to exceed 7 elements in the hierarchy, otherwise it will affect the consistency of the level. (3) If the number of experts interviewed is too large or the deviation of the selection criteria, it will affect the consistency of the analysis results. Shih, et al. [46] pointed out that the TOPSIS method, considering the benefit criterion or the cost criterion, can only reflect the relative proximity of the evaluation criteria within each assessment scheme, and does not reflect the relative proximity to the ideal optimal scheme. Shannon [47] suggested that if data can be obtained directly from the decision-making environment, the Entropy algorithm can be used to calculate the objective weight of the criterion. The more chaotic the measured value, the greater the chaos, the greater the weight of this criterion, the difference between the importance of different criteria, the method of dealing with uncertainty, and the reduction of subjective factors. CEGRA is that both qualitative and quantitative relationships can be identified among complex factors with insufficient information. The main feature of the CEGRA is that it can be applied with limited information, and can support an objective decision, based on different information. This method combines the advantages of AHP expert evaluation, entropy and TOPSIS, and can effectively deal with decision uncertainty, multi-criteria input and discrete data.”
Saaty, T. L. How to make a decision: The analytic hierarchy process. European Journal of Operational Research, 1971, 40, 9-10. Shih, H.S., Shyur, H.J., Lee, E.S. An extension of TOPSIS for group decision making, Mathematical and Computer Modelling. 2007, 45(7-8), 801–813. Shannon, C.E. A Mathematical Theory of Communication, The Bell System Technical Journal, 1948, 27(379-423), 623-656.

Round 3
Author Response
Review #2
I don't feel qualified to judge about the English language and style
Reply:
Thank you for your valuable comments. The paper has been shortened, modified, and undergone English language editing by MDPI.